# Targeting the Complement Cascade in the Pathophysiology of COVID-19 Disease

**DOI:** 10.3390/jcm10102188

**Published:** 2021-05-19

**Authors:** Nicole Ng, Charles A. Powell

**Affiliations:** Division of Pulmonary, Critical Care and Sleep Medicine, Icahn School of Medicine at Mount Sinai, New York, NY 10029, USA; nicole.ng2@mountsinai.edu

**Keywords:** COVID-19, SARS-CoV-2, complement pathway, lectin pathway, complement inhibitor, mannose-associated serine protease inhibitor

## Abstract

Severe coronavirus disease 2019 causes multi-organ dysfunction with significant morbidity and mortality. Mounting evidence implicates maladaptive over-activation of innate immune pathways such as the complement cascade as well as endothelial dysfunction as significant contributors to disease progression. We review the complement pathways, the effects of severe acute respiratory syndrome coronavirus 2 (SARS-CoV-2) on these pathways, and promising therapeutic targets in clinical trials.

## 1. Introduction

Severe coronavirus disease 2019 (COVID-19) is a multi-system disorder characterized by microvascular dysregulation and thrombosis that causes acute respiratory distress syndrome (ARDS), kidney injury, cardiomyopathy, and encephalitis. Evidence suggests that this is driven by exuberant host responses leading to maladaptive over-activation of innate immune pathways, most notably the complement cascade.

## 2. The Complement Cascade

The complement cascade is a host defense system that traces its biologic origins to over one billion years ago [1]. This system consists of more than fifty plasma proteins that interact to sense and respond to invasive pathogens [2]. It is a component of the innate immune response that links the innate and adaptive responses. Activation occurs via one of three pathways—classical, lectin, or alternative pathway, that converge to form C5 convertase, as well as numerous anaphylatoxins (Figure 1). C5 convertase generates C5b-9, also known as membrane attack complex (MAC). MAC creates cytotoxic membrane channels leading to cell death; however, even sublytic concentrations have significant immunomodulatory effects [3]. Anaphylatoxins produce a host of proinflammatory responses that range from increased vascular permeability to the activation and recruitment of leukocytes [4].

The important role of the complement cascade in responding to viral infections is demonstrated by evolutionary evidence of specific mechanisms employed by viruses to evade the complement system [5,6]. These include piracy of host regulators of complement activation (e.g., complement factor H by *Flaviviridae*), viral mimicry of host complement-related proteins (e.g., *Vaccinia viridae* encoding of vaccinia virus complement control protein, VCP), or encoding of proteins to prevent complement mediated effects (e.g., *Influenza A* encoding of matrix, M1, protein to prevent neutralization, or *Variola viridae* encoding smallpox inhibitor of complement enzyme, SPICE, to inhibit C3 convertase) [6,7,8,9].

The complement system’s role in innate immunity is an important first line of protection against invading pathogens. Importantly, excessive activation can lead to maladaptive host inflammatory responses with life-threatening effects. Recent research shows that exuberant complement activation causes pathogenic systemic inflammatory response syndrome, endothelial dysfunction, and multiorgan dysfunction [10,11,12].

## 3. Methodology and Literature Search

The PubMed literature search strategy (initial search date 14 August 2020 and updated on 14 April 2021) combined free-text keywords COVID, SARS-CoV-2, and complement. Filters for the search were applied, and the filters included humans [species], English [language], and journal articles [article types]. The number of search results totaled 618 items, including abstract only. ClinicalTrials.gov was accessed, the search strategy (initial search date 14 August 2020 and updated on 14 April 2021) utilized the search terms “COVID” for condition or disease, and “complement”. The total number of search results totaled 49. With the 667 items at hand, the cited items totaled 98, including 31 original articles, 41 reviews, 2 textbook, 16 clinical trials, and 8 websites.

## 4. Complement in SARS-CoV-1, MERS-CoV

The understanding of SARS-CoV-2 has leveraged prior research in SARS caused by related viruses in the beta-Coronaviridae family, SARS-CoV and Middle East respiratory syndrome coronavirus (MERS-CoV) [13]. SARS-CoV originated in Guangdong, China, in 2002, causing 8422 infections and 916 deaths, with a case fatality rate of 10.8% [14]. MERS-CoV originated in Saudi Arabia in 2012, causing 2468 infections and 851 deaths, with a case fatality rate of 34.4% [15]. SARS-CoV-2 emerged in December 2019 in Wuhan, China. By 17 April 2021, there were over 136.7 million confirmed cases and 3.0 million deaths, with a case fatality rate of 2.1% [16].

Gralinski and colleagues performed the first in vivo studies of SARS-CoV in C57BL/6J mice that showed complement activation as early as day 1 post-infection [11]. To determine whether this complement cascade response was protective or pathologic, they tested the impact of viral infection in C3-deficient (C3-/-) mice. C3-/- mice had equivalent viral loads as control mice, indicating that complement signaling is not necessary for SARS-CoV clearance in lung. Compared to controls, SARS-CoV-infected C3-/- mice exhibited less weight loss and respiratory dysfunction, as well as decreased inflammation as measured by neutrophils, monocytes, cytokines, and chemokines both on lung pathology as well as sera. These data suggest that SARS-CoV lung injury progression may be driven by complement activation.

Jiang and colleagues examined the role of complement in MERS-CoV in a model of complement activation driven by human dipeptidyl-peptidase 4 in transgenic (hDPP4-Tg) mice [12]. MERS-infected hDPP4-Tg mice were treated with anti-C5a receptor (C5aR) antibody or phosphate-buffered saline. Drug-treated mice had significantly lower lung viral titers, suggesting complement inhibition limits MERS-CoV replication and/or enhances clearance. Moreover, treated mice also exhibited less weight loss, lung and spleen damage, and inflammation measured by tissue macrophages and interferon-gamma, as well as decreased serum levels of several chemokines and cytokines. These data suggest that MERS-CoV infection results in dysregulated host immune response and that inhibition of the complement cascade may alleviate injury.

## 5. Complement in SARS-CoV-2

A common comorbidity in the clinical manifestation of severe COVID-19 pneumonia is the increased incidence of thrombotic events, including stroke and myocardial infarction, and generalized microvascular disfunction that is associated with shunt and microvascular thrombi [17,18,19,20]. These observations, along with prior research in SARS-CoV and MERS-CoV, indicate that the pathophysiology of SARS-CoV-2 involves the dysregulation of microvascular thromboinflammatory pathways, of which the complement cascade is most prominent [21].

The thromboinflammatory response is complex, requiring integration among the complement cascade and the coagulation and platelet pathways at multiple levels (Figure 2). For example, C5a and MAC induce the expression of tissue factor, MAC has direct effects on thrombin and platelet activation, and C5a enhances prothrombotic effects through the generation of IL-6, IL-8, and tumor necrosis factor-α [22,23]. These factors activate the endothelium and promote thrombin formation. Moreover, thrombin is able to cleave and activate C3 and C5, thus perpetuating the cycle of inflammation and coagulation. 

Of particular interest is the lectin pathway, which is activated through the binding of pathogen-associated molecular patterns (PAMPs) with the pattern recognition molecule, mannose-associated serine protease-1 (MASP-1) (Figure 3) [24,25]. MASP-1 in turn activates MASP-2, which is responsible for activating the rest of the lectin pathway [24,25]. MASP-1 has many substrates, including fibrinogen, factor XIIIa, prothrombin, and thrombin-activatable fibrinolysis inhibitor (TAFI), all of which act in concert to promote clot formation (Figure 3) [24]. Clinically, the lectin pathway is known to induce microvascular endothelial cell (MVEC) injury, as seen in thrombotic microangiopathies. Elhadad and colleagues found that inhibition of MASP-2 in human MVEC reduced cell injury measured by reduction in caspase-8 (apoptotic) activity [26].

Yu and colleagues showed that SARS-CoV-2 spike protein directly activates the alternative pathway on cell surfaces and that alternative pathway factor D inhibitor blocks SARS-CoV-2 mediated complement activation [27]. Gao and colleagues showed that the N protein of MERS-CoV, SARS-CoV, and SARS-CoV-2 interacts with and potentiates the mannose-binding lectin pathway of the complement cascade via MASP2 [28]. Blockade of this interaction, either by MASP-2 inhibition or by genetic deletion, led to attenuation of lung injury and increased survival in mice.

In patients with severe COVID-19 pneumonia, complement cascade over-activation has been well demonstrated. C5a levels (measured in plasma and bronchoalveolar lavage) were significantly higher in patients with severe COVID-19 pneumonia compared with those with mild disease [28,29]. High-throughput transcriptomic and proteomic studies have also corroborated these findings in the sera/plasma of COVID-19 patients. In a study by Shen and colleagues, proteomic and metabolomic profiling of sera showed general upregulation of complement system proteins, including MAC proteins such as C5, C6, and C8 [30]. Messner and colleagues similarly found consistent activation of the classical complement pathway (e.g., C1R, C1S), alternative pathway (e.g., factor B), and complement modulators (e.g., factor I, factor H) [31]. Additionally, D’Alessandro and colleagues found elevated levels of interleukin-6 that correlated with significant dysregulation in the serum levels of coagulation factors, antifibrinolytic components, and upregulation of complement activity [32].

Numerous histopathologic studies have also confirmed the presence of complement activation not only in the lungs but also in other organs such as kidneys [33]. Lung tissue from post-mortem SARS-CoV-2 autopsy patients demonstrated strong immunohistochemical expression of mannose-binding lectin (MBL), MASP-2, C3, C4a, C4d, and C5b-9 proteins [20,28,29]. 

In a study by Rendeiro and colleagues, spatially resolved single-cell analyses of post-mortem lung tissue from patients with COVID-19 showed significant reduction in alveolar lacunar spaces, as well as increased immune cell infiltration and apoptotic activity. Immunohistochemical data showed the upregulation of inflammatory pathways (e.g., interferon and interleukin signaling) in early COVID-19 (<14 days), whereas the upregulation of coagulation, complement (e.g., C5b-9 staining), and apoptosis (e.g., Caspase-3 staining) pathways were seen in late COVID-19 (>30 days). In general, higher levels of inflammation, macrophage infiltration, complement activation, and fibrosis were found to be specific to severe, late COVID-19 [34]. Taken together, these findings suggest early disease is mediated by innate inflammatory responses and late severe disease by pathogen-independent mechanisms driven by exuberant thromboinflammatory responses and complement activation.

Importantly, complement activation in COVID-19 was associated with clinical outcomes. Across multiple studies by Cugno et al., de Nooijer et al., Holter et al., and Peffault de Latour et al., complement factors and soluble terminal complement complex (sC5b-9 or sMAC) levels were associated with either increased odds of respiratory failure, need for oxygen therapy, and/or more severe disease [35,36,37,38].

Complement activation not only leads to a maladaptive immune responses but also to microvascular injury and thrombosis [20]. These endothelial manifestations have been reviewed in several studies [39,40,41,42]. Increasingly, histopathologic studies have shown evidence of widespread endothelial dysfunction, hypercoagulability, and imbalance of innate and adaptive immune responses [43,44]. 

Activated leukocytes upregulate tissue factor and generate neutrophil extracellular traps, which interact with von-Willebrand factor, platelets, and factor XIII to perpetuate immunothrombosis [45].

Endothelial cells are directly infected by SARS-CoV-2 through the angiotensin-converting enzyme 2 (ACE-2) receptor [46]. This invasion leads to the release of damage associated molecule patterns (DAMPs), promoting prothrombotic and pro-inflammatory mediators, including continued activation of the complement cascade [47,48]. DiNicolantonio and colleagues have hypothesized that SARS-CoV-2 induce endothelial cells to express tissue factor, promoting these thromboinflammatory effects [49]. This was studied by Rosell and colleagues, who showed that circulating tissue factor-positive extracellular vesicles levels are higher in patients with COVID-19, and also were correlated with disease severity and mortality [50]. Gavriilaki and colleagues noted that the clinical manifestations of endothelial dysfunction are extensive and include microvascular lung thrombosis, venous thrombosis (e.g., pulmonary embolism, deep vein thrombosis, cerebral venous thrombosis), arterial thrombosis (e.g., cardiovascular or cerebral vascular disease, acute limb ischemia), renal disease, and neurological complications (e.g., encephalopathy) [51]. As such, antithrombotic drugs have been proposed as potential adjunctive therapies for the prevention of COVID-19-associated thrombosis [52].

Concerted sustained immune, complement, and hemostatic responses lead to clinical manifestations that include abnormal inflammatory (e.g., C-reactive protein, ferritin, interleukin-6) and coagulation (e.g., d-dimer, lactate dehydrogenase) markers to the multiorgan thrombotic complications [45,53,54]. The culmination of these responses, if undeterred, is acute respiratory distress syndrome and multiorgan failure, with a high mortality [55].

## 6. Targeted Complement Therapies in COVID-19 Pneumonia

Several clinical trials are evaluating therapies targeting the complement pathway in the treatment of patients with COVID-19 pneumonia. These trials as well case reports, case series and observational cohorts that have been published are listed in Table 1. These drugs target different parts of the cascade—C1 esterase, C3, C5a, C5aR, C5, or MASP-2, with each having different implications (Figure 4). 

C1 esterase is the initiating enzyme specific to the classical complement pathway. The C1 esterase inhibitor is a plasma serine protease inhibitor that not only inactivates C1 esterase but also has regulatory effects on MASPs, coagulation factors, as well as kallikrein [79]. The latter is of particular interest, as kallikrein cleaves kininogen to form bradykinin, which has also been postulated to mediate disease in COVID-19 [56,80,81]. The C1 esterase inhibitor conestat alfa (Pharming; Warren, NJ, USA), currently used for hereditary angioedema, is now being studied in Phase II trials for COVID-19 pneumonia [57,58].

C3 activation, positioned upstream, is involved in each of the three complement pathways. C3 inhibition was shown to have greater reduction in C3 and C5b-9 levels compared to C5 inhibition, suggesting a more potent anti-inflammatory effect [82,83]. Mastaglio and colleagues were the first to treat a COVID-19 pneumonia patient with the C3 inhibitor, AMY-101 (Amyndas Pharmaceuticals; Philadelphia, PA, US) [59]. AMY-101 is a small peptide compound that binds C3, preventing its binding to and cleavage by C3 convertase [84,85]. AMY-101 and a similar investigational C3 inhibitor, APL-9 (Apellis Pharmaceuticals; Crestwood, KY, US), are in Phase II and Phase I and II trials, respectively, for COVID-19 pneumonia [60,61].

The C5-C5aR axis is an attractive therapeutic target because this complex generates the potent anaphylatoxin C5a and MAC [86]. Within this axis, compounds that specifically target C5a or C5aR allow for the selective blockade of pro-inflammatory C5a effects while maintaining the killing activity of MAC. The observed hyperactivation of C5a in COVID-19 suggests that this may be a promising target [87]. There are two recombinant C5a antibodies currently being studied in open-label and randomized-controlled trials for COVID-19 pneumonia, BDB-001 (Staidson; Beijing, China) and vilobelimab (InflaRx; Jena, Germany), respectively [28,62,63,64,88]. The first C5aR antibody, avdoralimab (Innate Pharma; Marseilles, France), currently undergoing phase I trials for solid tumors is in phase II trials for severe COVID-19 pneumonia in France [29,65].

C5 inhibition in patients with paroxysmal nocturnal hematuria (the mainstay of treatment in this complementopathy) led to milder COVID-19 disease course than those untreated [89]. Treated patients did not require hospitalization or supplemental oxygen, and had lower levels of inflammatory markers, suggesting that early complement blockade may be protective in COVID-19. C5 inhibitors include eculizumab (Alexion Pharmaceuticals; Boston, MA, USA) and ravulizumab (Alexion Pharmaceuticals; Boston, MA, USA), with the latter having a longer half-life. Eculizumab was the first complement inhibitor studied; it was given US Food and Drug Administration (FDA) approval in 2007 for paroxysmal nocturnal hematuria [90]. Since then, its use has expanded to not only other complementopathies but has also paved the way for the various anti-complement therapies discussed herein [91]. After a phase II trial of eculizumab in the US was shown to demonstrate efficacy in patients with COVID-19 pneumonia, a phase III randomized controlled trial of ravulizumab was initiated in 39 institutions, including our own [66,71,72,73,74] Other C5 inhibitors include zilucoplan (UCB; Brussels, Belgium) and tesidolumab (Novartis and Alcon; Fort Worth, TX, USA), both undergoing phase II trials for myasthenia gravis and macular degeneration, respectively, are also being studied in COVID-19 in Belgium and the United Kingdom [75,76,92].

More recently, MASP inhibition has been considered in the therapeutic armamentarium against COVID-19. Eriksson and colleagues found that complement activation occurs through the MBL pathway, and that elevated MBL plasma levels were associated with thromboembolic events and disease severity [93]. The only clinically available MASP inhibitor is narsoplimab (Omeros and Lonza; Basel, CH), targeting MASP2, and was granted FDA Breakthrough Therapy and Organ Drug designations for thrombotic microangiopathies [94]. Rambaldi and colleagues studied the first lectin-pathway inhibitor treatment with narsoplimab in six patients [77].

Several of these inhibitors have preliminarily reported promising results in compassionate use and early phase clinical trials for COVID-19 pneumonia [28,29,59,63,66,67,68,75,77]. Of note, a phase 3 clinical trial with ravulizumab in severe COVID-19 patients stopped enrollment on 13 January 2021 after a prespecified interim analysis determined futility [95]. It is important to emphasize that the inclusion criteria included patients with severe disease, defined as requiring non-invasive ventilation (e.g., BiPAP) or mechanical ventilation. It must be considered that C5 inhibition by this point may be too late—activation of the complement cascade as well as other inflammatory pathways have likely already overwhelmed. As such, ravulizumab (and likely all immunomodulatory therapies) may need to be given earlier in COVID-19 disease course to render benefit. Two studies are ongoing to investigate ravulizumab in earlier COVID-19 disease in the United States and the United Kingdom [73,74]. The final results of these trials are awaited.

## 7. Conclusions

The clinical manifestations of COVID-19 are wide and unpredictable; they range from asymptomatic or short-lived illness to fatal multi-organ system failure [96]. This underlies the importance of understanding the pathophysiology of disease in order to characterize, prognosticate, and treat patients who are at risk for severe disease.

The complement cascade is of special interest, as it is uniquely poised to address both the hyperinflammation and hypercoagulability that are characteristic of severe COVID-19 disease [23,43,44,97]. Recent research has shown that overactivation of complement drives the severity of SARS infections and that inhibition of the complement pathway reduces disease burden [11,12]. Complement laboratory values in patients with COVID-19 pneumonia suggest variability in complement activation of specific pathways, such that an individualized approach to treatment may be feasible.

The optimal complement pathway component inhibition target is also not yet clear, even in the more studied complementopathies. Ongoing trials in SARS-CoV-2 target complement inhibition at the level of C1 esterase, C3, C5, C5a, C5aR, or MASP. To address hyperinflammation, it is theorized that upstream targets (e.g., C3 inhibition) would yield the greatest anti-inflammatory effects [59]. However, this may come at the cost of an increased risk of infection, thus some proponents favor C5a or C5aR over C5 inhibition due to the retained ability to generate C5b-9 or MAC [29,98]. Alternatively, C1 esterase inhibition with its kallikrein effects has also been proposed as a promising target to address the elevated bradykinin levels caused by dysregulated ACE2 activity in SARS-CoV-2 [81]. 

With regards to hypercoagulability, the complement inhibitor compounds in active COVID-19 clinical trials likely have minimal effects; these primarily target the expression of tissue factor and plasminogen activator inhibitor-1 [24]. The potential impact of the coagulation cascade may be higher for MASP inhibitors because of interaction with fibrinogen, factor XIIIa, prothrombin, and TAFI [24]. Particularly, MASP-1 is also the initiating protease that recognizes PAMPs, leading to the activation of the lectin pathway in the complement cascade [25]. Currently, only the MASP-2 inhibitor, narsoplimab, is being studied for thrombotic microangiopathies [77]. Further studies are necessary to investigate the role of MASP inhibitors in SARS-CoV-2.

Taken together, SARS-CoV-2 causes severe multi-system thromboinflammatory disease driven by over-activation of inflammatory cascades. Recent preclinical and clinical studies show that targeting the complement cascade can impact the pathophysiologic derangements driven by over-activation of the complement cascade. Further research is necessary to better refine the targeting approaches and to achieve improved outcomes for patients with COVID-19 respiratory disease. 

## Figures and Tables

**Figure 1 jcm-10-02188-f001:**
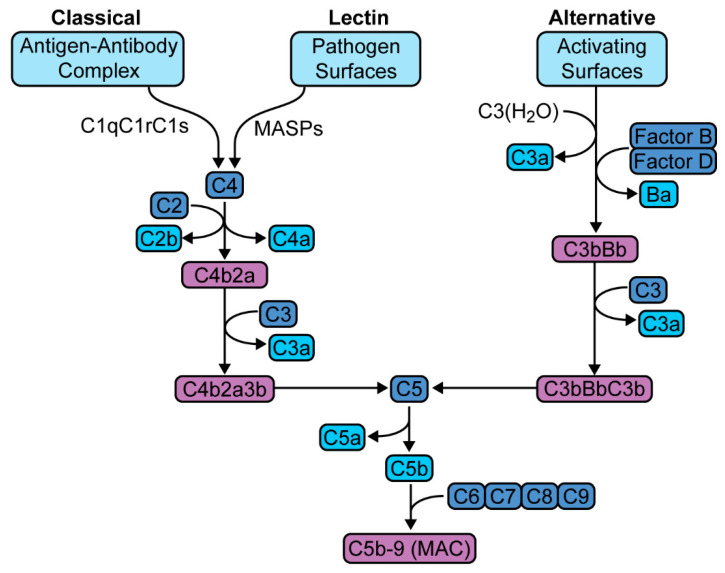
Complement activation pathways. The classical pathway is activated by the binding of antibody-antigen complex with the C1 esterase complex (C1qC1rC1s), which then forms C3 convertase (C4b2a). The lectin pathway is initiated by the binding of mannose-binding lectin with mannose residues on pathogen surfaces, which activates mannose-associated serine proteases (MASPs) to form the same C3 convertase (C4b2a). The alternative pathway is triggered directly by antigen and also through spontaneous autoactivation, leading to the formation of a similar C3 convertase (C3bBb). These C3 convertases hydrolyze C3 to yield the C5 convertases (C4b2a3b and C3bBbC3b, respectively), which culminates in the generation of C5b-9 or MAC (membrane attack complex).

**Figure 2 jcm-10-02188-f002:**
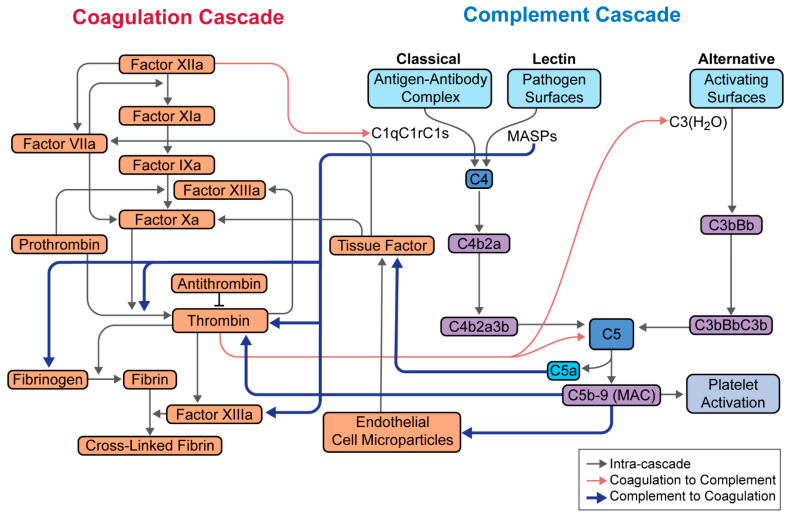
Integration between the complement and coagulation cascades. Relationship between the complement and coagulation cascades, as well as platelet activation is highlighted. Most notable are the effects of MASPs on thrombin, factor XIIIa, and fibrinogen; C5a on tissue factor; and C5b-9 on thrombin. Adapted from Hill et al. [23].

**Figure 3 jcm-10-02188-f003:**
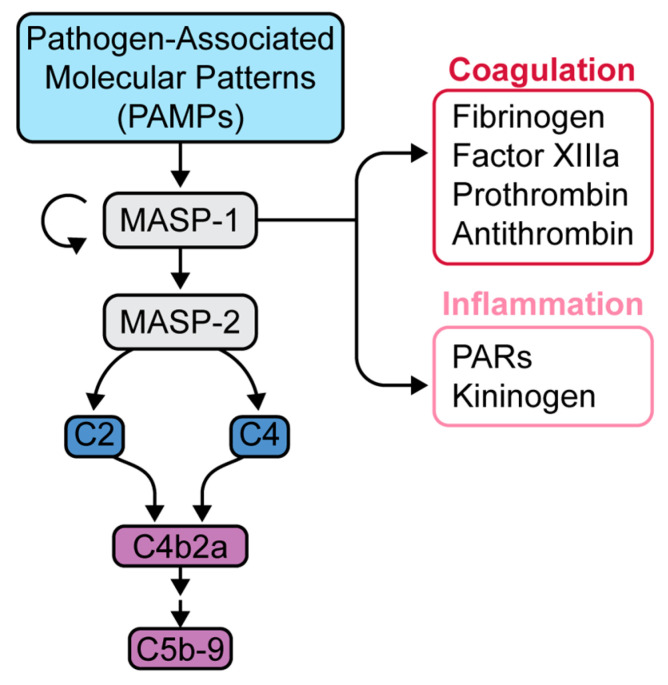
Mannose-binding lectin pathway and downstream effects. Initiation of the lectin pathway leads to the activation of MASP-1, which is capable of further autoactivation as well as activation of MASP-2. MASP-2 is responsible for the generation of subsequent anaphylatoxins and MAC. MASP-1 has much more relaxed substrate specificity, rendering additional effects in the coagulation and inflammatory cascades. Adapted from Dobo et al. [25].

**Figure 4 jcm-10-02188-f004:**
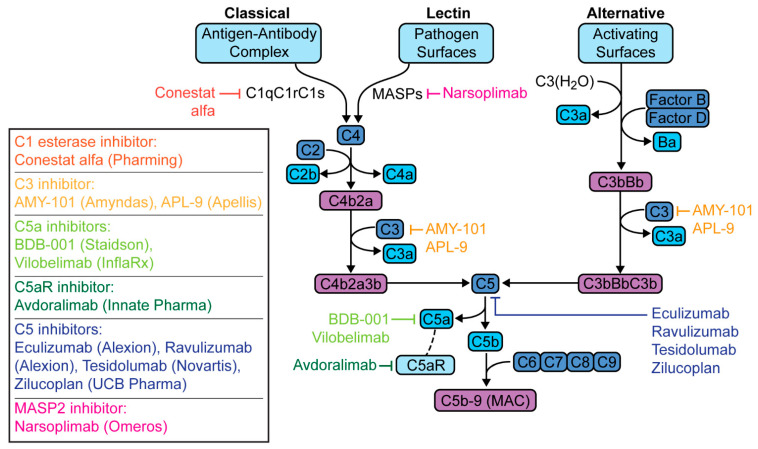
Therapeutic targets in the complement cascade for the treatment of coronavirus disease 2019 (COVID-19) pneumonia.

**Table 1 jcm-10-02188-t001:** Therapeutic targets in the complement cascade for the treatment of COVID-19 pneumonia. * Nonrandomized, controlled cohort study. Abbreviations: NCT: National Clinical Trial. Country Abbreviations: BD: Bangladesh; BE: Belgium; BR: Brazil; CH: Switzerland; CN: China; DE: Germany; ES: Spain; FR: France; GB: United Kingdom; ID: Indonesia; IN: India; IT: Italy; MX: Mexico; NL: Netherlands; PE: Peru; RU: Russia; US: the United States.

Drug	Sponsor	Study Type	Status	Location	NCT	Publications
**C1 esterase inhibitor**						
Conestat-alfa	Pharming	Case series (*n* = 5)	Not available	CH		Urwyler [56]
		Phase 2 (*n* = 120)	Recruiting	BR, CH, MX	04414631 [57]	
Ruconest^®^	Pharming	Phase 2 (*n* = 120)	Recruiting	US	04530136 [58]	
**C3 inhibitor**						
AMY-101	Amyndas	Case report (*n* = 1)	Not available	IT		Mastaglio [59]
		Phase 2 (*n* = 144)	Not yet recruiting	Not listed	04395456 [60]	
APL-9	Apellis	Phase 1,2 (*n* = 66)	Recruiting	US, BR	04402060 [61]	
**C5a inhibitor**						
BDB-001	Staidson	Case series (*n* = 2)	Not available	CN		Gao [28]
		Phase 2,3 (*n* = 368)	Recruiting	BD, CN, ES, ID, IN	04449588 [62]	
Vilobelimab	InflaRx	Phase 2 (*n* = 30)	Not available	NL		Vlaar [63]
		Phase 2,3 (*n* = 390)	Recruiting	BE, BR, FR, DE, NL, PE, RU	04333420 [64]	
**C5aR inhibitor**						
Avdoralimab	Innate	Phase 2 (*n* = 208)	Active, not recruiting	FR	04371367 [65]	
**C5 inhibitor**						
Eculizumab	Alexion	Cohort * (*n* = 80)	Not available	FR		Annane [66]
		Case series (*n* = 4)	Not available	IT		Diurno [67]
		Case series (*n* = 3)	Not available	US		Laurence [68]
		Phase 2 (*n* = 120)	Recruiting	FR	04346797 [69]	
		Expanded Access	Available	Not listed	04288713 [70]	
		Expanded Access	Not available	FR, US	04355494 [71]	
Ravulizumab	Alexion	Phase 3 (*n* = 270)	Active, not recruiting	ES, FR, GB, JP, US	04369469 [72]	
		Phase 3 (*n* = 32)	Recruiting	US	04570397 [73]	
		Phase 4 (*n* = 1167)	Recruiting	GB	04390464 [74]	
Tesidolumab	Novartis	Case series (*n* = 5)	Not available	GB		Zelek [75]
Zilucoplan	UCB	Phase 2 (*n* = 81)	Active, not recruiting	BE	04382755 [76]	
**MASP-2 inhibitor**						
Narsoplimab	Omeros	Case series (*n* = 6)	Not available	IT		Rambaldi [77]
		Phase 2, Adaptive (*n* = 1500)	Recruiting	US	04488081 [78]	

## Data Availability

No new data were created or analyzed in this study. Data sharing is not applicable to this article.

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
