# Peer review of "Targeting the Complement Cascade in the Pathophysiology of COVID-19 Disease"

_jcm, 2021, doi:10.3390/jcm10102188_

Round 1

Reviewer 1 Report

The ms reports a review on the possible involvement of the complement system in COVID-19. Authors suggest that activation of the complement cascade by SARS-CoV-2 infection may support the use of drugs blocking complement activation.

Main comments

  1. There is no information how the papers discussed in the review were selected and what kind of methodology has been used (meta-analysis etc).
  2. Several published papers on complement activation in COVID-19 were missed. In particular, most of them were including series of patients larger than those reported in the quoted papers. I believe that all the papers should be reviewed and quoted in a more appropriate manner.
  3. It has been suggested that complement activation and in particular C5b-9 may be responsible for endothelial perturbation in COVID-19 (Cugno et al J Autoimmun 2021). I think that this aspect should be discussed as well.

Reviewer 2 Report

In this article, the authors review the most recent evidence on complement activation and COVID-19, particularly from a therapeutic perspective. The manuscript is well written and organized; the figures are clear and informative. Although COVID-19 is a new pathological entity, its relation with complement has already been extensively investigated, and recent good reviews are available on the subject (Java et al. JCI insight 2020, Noris et al. Kidney Int. 2020, Lo et al. J Immunolol. 2020). For this reason, the novelty of the manuscript is partially reduced. However, considering the rapid accumulation of evidence in this field and the specific therapeutic angle here discussed, this can still be regarded as an interesting and informative review.

Here are few comments:

Considering the title “Targeting the Complement Cascade in the Pathophysiology of COVID-19 Disease,” the authors may consider expanding section 4, “rationale for complement in SARS-CoV-2,” by elaborating further on COVID-19 pathophysiology. In particular, how SARS-Co2V-2 infection relates to microvascular thrombo-inflammation and complement activation. 

In Section 5, “Complement in SARS-CoV-2,” the authors may consider discussing other significant recent studies on the subject such as (Holter et al. PNAS 2020; Kulasekararaj et al. Br J Hematol 2020; Mastellos et al. Clin Immunol. 2020).

Author Response

Response to Reviewer 2Comments

In this article, the authors review the most recent evidence on complement activation and COVID-19, particularly from a therapeutic perspective. The manuscript is well written and organized; the figures are clear and informative. Although COVID-19 is a newpathological entity, its relation with complement has already been extensively investigated, and recent good reviews are available on the subject (Java et al. JCI insight 2020, Noris et al. Kidney Int. 2020, Lo et al. J Immunolol. 2020). For this reason, the novelty of the manuscript is partially reduced. However, considering the rapid accumulation of evidence in this field and the specific therapeutic angle here discussed, this can still be regarded as an interesting and informative review.

Thank you for the comments. We are pleased that you agree with the review’s therapeutic angle and its importancefor the field.

Below please find our response to each comment:

Point 1:

Considering the title “Targeting the Complement Cascade in the Pathophysiology of COVID-19 Disease,” the authors may consider expanding section 4, “rationale for complement in SARS-CoV-2,” by elaborating further on COVID-19 pathophysiology. In particular, how SARS-Co2V-2 infection relates to microvascular thrombo-inflammation and complement activation.

Response 1:Thank you for this suggestion.We have combined the prior sections 4 and 5 into a single,more comprehensivesection (e.g. Section 5),which can be found beginning line 98.

We have updated our search and included the following references to elaborate on the evidence for complement activation.-Reference 30: Shen et al Cell 2020 (lines 152-155)

-Reference 31: Messner et al Cell Syst 2020 (lines 155-157)

-Reference 32: D’Alessandro et al J Proteome Res 2020 (lines 157-160)

-Reference 33: Pfister et al Frontiers Immunol 2020 (lines 161-162)

-Reference 34: Rendeiro et al Nature 2021 (lines 166-176)

-Reference 35: Holter et al PNAS 2020 (lines 177-181)

-Reference 36: Cugno et al J Autoimmun 2021 (lines 177-181)

-Reference 37: de Nooijer et al, J Infect Dis 2021 (lines 177-181)

-Reference 38: Peffault de Latour et al Haematologica 2020 (lines 177-181)

Point 2:In Section 5, “Complement in SARS-CoV-2,” the authors may consider discussing other significant recentstudies on the subject such as (Holter et al. PNAS 2020; Kulasekararaj et al. Br J Hematol 2020; Mastellos et al. Clin Immunol. 2020).

Response 2:Thank you for these suggestions, we have included additional references on this new Section 5.

-Reference 30: Shen et al Cell 2020 (lines 152-155)-Reference 31: Messner et al Cell Syst 2020 (lines 155-157)

-Reference 32: D’Alessandro et al J Proteome Res 2020 (lines 157-160)-Reference 33: Pfister et al Frontiers Immunol 2020 (lines 161-162)

-Reference 34: Rendeiro et al Nature 2021 (lines 166-176)

-Reference 35: Holter et al PNAS 2020 (lines 177-181)-Reference 36: Cugno et al J Autoimmun 2021 (lines 177-181)

-Reference 37: de Nooijer et al, J Infect Dis 2021 (lines 177-181)

-Reference 38: Peffault de Latour et al Haematologica 2020 (lines 177-181)

-Reference 46: Perico et al Neprology 2021 (lines 190-191)

-Reference 47: Conway et al Haemost 2020 (lines 191-193)

-Reference 48: Manolis et al J Cardiovasc Pharmacol Therapeutics 2020 (lines 191-193)

-Reference 51: Gavriilaki et al Curr Hypertens Rep 2020 (lines 198-202)

-Reference 52: McFadyen et al Circ Res 2020 (lines 203-204)

-Reference 45: Jayarangaiah et al Clin Appl Thromb Hemost 2020 (lines 205-208)

-Reference 53: Varghese et al Immunobiology 2020 (lines 205-208)

-Reference 54: Chau et al Arthritis Rheumatol 2021 (lines 205-208)

-Reference 55: Lippi et al Annals Translational Med 2020 (lines 208-210)

-Reference 72: Mastellos et al Clin Immunol 2020 (lines 238-239)

-Reference 73: Mishra et al Neuroimmunomodulation 2020 (lines 238-239)

-Reference 84: Kulasekararaj et al Br J Hematol (lines254-256)
